# Germline Genetic Findings Which May Impact Therapeutic Decisions in Families with a Presumed Predisposition for Hereditary Breast and Ovarian Cancer

**DOI:** 10.3390/cancers12082151

**Published:** 2020-08-03

**Authors:** Carolina Velázquez, De Leeneer K., Eva M. Esteban-Cardeñosa, Francisco Avila Cobos, Enrique Lastra, Luis E. Abella, Virginia de la Cruz, Carmen D. Lobatón, Kathleen B. Claes, Mercedes Durán, Mar Infante

**Affiliations:** 1Cancer Genetics Group, Instituto de Biología y Genética Molecular (UVa-CSIC), 47003 Valladolid, Spain; carolina-velazquez@hotmail.com (C.V.); Eva.Maria.Esteban.Cardenosa@sergas.es (E.M.E.-C.); clobaton@ibgm.uva.es (C.D.L.); merche@ibgm.uva.es (M.D.); 2Center for Medical Genetics, Ghent University Hospital, and Cancer Research Institute Ghent (CRIG), Ghent University, 9000 Ghent, Belgium; kim.deleeneer@ugent.be (D.L.K.); francisco.avilacobos@ugent.be (F.A.C.); kathleen.claes@ugent.be (K.B.C.); 3Unit of Genetic Counseling in Cancer, Complejo Hospitalario de Burgos, 09006 Burgos, Spain; elastra@saludcastillayleon.es; 4Unit of Genetic Counseling in Cancer, Hospital Universitario Rio Hortega, 47012 Valladolid, Spain; leabella@saludcastillayleon.es (L.E.A.); soyvicky_8@hotmail.com (V.d.l.C.)

**Keywords:** HBOC, genetic testing, variant prioritisation, therapeutic selection, family history, clinical guidelines

## Abstract

In this study, we aim to gain insight in the germline mutation spectrum of *ATM*, *BARD1*, *BRIP1*, *ERCC4*, *PALB2*, *RAD51C* and *RAD51D* in breast and ovarian cancer families from Spain. We have selected 180 index cases in whom a germline mutation in BRCA1 and BRCA2 was previously ruled out. The importance of disease-causing variants in these genes lies in the fact that they may have possible therapeutic implications according to clinical guidelines. All variants were assessed by combined annotation dependent depletion (CADD) for scoring their deleteriousness. In addition, we used the cancer genome interpreter to explore the implications of some variants in drug response. Finally, we compiled and evaluated the family history to assess whether carrying a pathogenic mutation was associated with age at diagnosis, tumour diversity of the pedigree and total number of cancer cases in the family. Eight unequivocal pathogenic mutations were found and another fourteen were prioritized as possible causal variants. Some of these molecular results could contribute to cancer diagnosis, treatment selection and prevention. We found a statistically significant association between tumour diversity in the family and carrying a variant with a high score predicting pathogenicity (*p* = 0.0003).

## 1. Introduction

Genetic testing is a key tool for cancer prevention through family history analysis and the identification of inherited mutations. Thus, testing for individuals at risk of hereditary breast and ovarian cancer (HBOC) is encouraged to implement control and prevention measures through genetic counselling [1]. Uncovering the genetic causes of HBOC cases enables early detection and prevention. Due to its inheritable nature, family members may benefit from genetic counselling [2,3].

Although hereditary cancer accounts for a small proportion of the total cases, in absolute numbers they represent a significant number of mutation carriers who can benefit from clinical management in order to reduce morbidity and mortality [4].

After screening for BRCA1/2 genes, the underlying genetic predisposition remains elusive in a proportion of high risk cancer cases. Therefore, we undertook screening of other genes in a cohort of patients with a presumed genetic predisposition for breast and ovarian cancer, in whom germline mutations in BRCA1 and BRCA2 were previously ruled out.

DNA repair defects prompt genetic instability, a well-known carcinogenic trigger. Therefore, functional defects in any DNA repair gene can contribute to the accumulation of errors and, ultimately, to cellular transformation. In fact, most HBOC susceptibility genes encode proteins which are involved in DNA repair. Mutations in these genes compromise the effectiveness of DNA repair, leading to genomic instability. Reciprocally, the accumulation of these mutations would be promoted by genomic instability, and this mutagenic environment would eventually enhance tumour progression. Proteins, such as ATM, BARD1, and Fanconi Anemia (FA) BRIP1, ERCC4, PALB2, RAD51C and RAD51D proteins, are involved in DNA repair pathways and have previously been linked to HBOC susceptibility associated with low to moderate risks [5,6].

Exploiting DNA repair defects of the mutated cells has emerged as a targeted therapy strategy. Germline mutations in HBOC susceptibility genes not only confer cancer risk to carriers, but may also determine the response to certain treatment options, providing the opportunity to customise chemotherapy [7,8].

In Spain, particularly in our region, the standard practice for patients at high risk for HBOC is to screen germline mutations in BRCA1 and BRCA2 genes. Consequently, the purpose of this work is to investigate whether by increasing the number of genes tested (some BRCA-FA pathway genes involved in HBOC susceptibility and *ATM* and *BARD1* genes), more families could benefit from prevention measures according to SEOM Clinical Guidelines [3]. Furthermore, it aims to be a first approach to select other clinically useful genes for better management in our BRCA negative families; and, hence, to implement a personalised panel for families that meet high-risk criteria in our target population.

## 2. Results

### 2.1. Mutation Screening

All 180 individuals included in this study were previously screened for BRCA1/2 point mutations and large rearrangements, with negative results. We previously published a *PALB2* deleterious mutation that was identified in this cohort [9].

To further characterise this 180 index-case cohort, we selected some BRCA-FA pathway genes first. Therefore, seven additional genes implicated in HBOC predisposition and DNA damage response were analysed: *ATM*, *BARD1, BRIP1, ERCC4, PALB2, RAD51C* and *RAD51D*. All the variants identified in the cohort are listed in Appendix A.

We found eight pathogenic variants in 180 unrelated probands (4.4%): c.3663G>A (p.Trp1221*), c.4776+2T>C and c.8934_8935delTG (p.Glu2979Alafs*9) in *ATM*; c.484C>T (p.Arg162*) in *BRIP1*; c.1251T>A (p.Cys417*) and c.584+1G>A in *ERCC4*; c.1857delT (p.Phe619Leufs*9) in *PALB2* and c.94_95delGT (p.Val32Phefs*38) in *RAD51D*. All mutations have already been reported in disease databases (ClinVar, dbSNP, LOVD) or in literature [9,10]. The personal and family history for each pathogenic variant is described in Table 1.

Cancer types are abbreviated as following: Br, breast; BilBr, bilateral breast; Ov, ovarian; Gas, gastric; Col, colon; Pan, pancreas; Thy, thyroid; H&N, head and neck; Pr, prostrate; Cvx, cervix; Ute, uterus; Kid, kidney; CUP, cancer of unknown primary origin. The age of diagnosis is indicated in parentheses, n.a not available data.

### 2.2. Segregation Analysis

Segregation analysis was performed for some *BRIP1*, *RAD51D* and *ATM* pathogenic mutations. Pedigrees showing the analysis results are displayed in Figure 1. Segregation with the cancer phenotype in certain variants was incomplete in some families: such as the family with c.94_95delGT *RAD51D* mutation (Figure 1b) and the family with c.3663G>A *ATM* variant (Figure 1c). For the family with c.8934_8935delTG variant in *ATM* (Figure 1d) a first degree cancer affected relative (Figure 1d, III.12) turned out to be negative.

Several additional variants possibly impacting the function of the protein were identified in our cohort. These can be found above the corresponding protein schemes (Figure 2).

### 2.3. Variant Prioritisation

In order to prioritise the filtered variants, both in silico functional assessment using the CADD algorithm (Figure 3) and a review of the possible implications of any mutations in drug response were conducted. The higher the CADD score, the greater the likelihood that this variant will be deleterious. A CADD score >25 was previously suggested as cut-off for of a damaging effect. Indeed, all pathogenic mutations surpassed that CADD score (33.6 ± 5.8). After the analysis, a total of 22 variants presented with a CADD score >25 (“high-CADD score” onwards) (Figure 3).

Interestingly, for two putative splicing variants, CADD analysis returned a high score: one of them, c.584+1G>A in *ERCC4* was previously tested by our group at cDNA level resulting in a exon 3 skipping: p.Ile131Serfs*2 [11]. The other one, c.4776+2T>C in *ATM*, has been previously analysed in fibroblasts derived from a homozygous carrier [12]. In this work, we investigated the effect of c.4776+2T>C splicing variant in cDNA. Thus, we performed a reverse transcription of the patient RNA. An amplicon spanning exons 29 to 33 was amplified. The mutation carrier transcript revealed two bands: a 769 base-pair (bp) band corresponding to the expected wild-type fragment and an aberrant transcript band (604 bp), while controls only showed the wild-type band (Figure 4A). The subsequent sequence of the 604 bp transcript revealed exon 31 skipping (r.4612_4776del165; p.Val1538_Glu1592del55) (Figure 4).

### 2.4. Therapeutic Implications

In order to contribute to the clinical utility, we took into consideration the high-CADD score variants considered by the cancer genome interpreter as genomic biomarkers of drug response with different levels of clinical relevance. Hypothetically, alterations in some actionable cancer genes could determine the treatment effect. Indeed, the precision oncology concept is based on selecting the chemotherapy regimen according to the supposed implications of the mutational profile. The different drugs for each gene are detailed in Figure 3.

For some of the mutations, drug responses are predicted. This information could be taken into consideration to select some of the available inhibitors for cancer treatment. In that sense, as a potential but inconclusive valuable information, seven patients harbouring suggested pathogenic variants could benefit from targeted therapy options with ATR inhibitors, PARP1 inhibitors and DNA-PKc inhibitors. In line with this targeted therapy based on synthetic lethality concept, some of the *PALB2* variants (p.Phe619Leufs*9, p.Val932Met and p.Leu939Trp) and the p.Thr287Ala variant in *RAD51C* could be determining a response to PARP inhibitors. For the therapy with the traditional drugs (cisplatin and other platinum compounds), we could expect a response in the *ERCC4* and *PALB2* mutation carriers.

### 2.5. Genotype-Phenotypic Correlations

We evaluated if high-CADD score variants (Figure 3) were associated with younger age at diagnosis and a more diverse tumour spectrum.

We first compared the mean age at diagnosis between high-CADD score mutation carriers and non-carriers (one-sided *t*-test *p*-value = 0.1673). These analyses showed, at a 5% level of statistical significance, that carrying a variant with a high-CADD score was not associated with an earlier tumour onset (mean age of diagnosis is the same in both groups).

Secondly, we checked whether carriers of a high-CADD score mutation showed greater tumour-type diversity in the family (Figure 5). Pearson’s chi-squared test, with Yates’ continuity correction, suggested an association between tumour diversity in the family, and the carrying of a variant with a high CADD score (5% level of statistical significance (*p* = 0.0003)). In addition, high-CADD score mutation carriers presented a higher number of cancer cases in the pedigree than negative cases (6.4 vs 4.4 mean number of cases respectively); therefore, the condition of carrying a high-CADD score mutation is associated with an increased burden of cancer cases in the family (*p*-value = 0.0023).

## 3. Discussion

The clinical management of HBOC families makes cancer prevention possible through the identification of individuals at high-risk. High penetrance genes explain a small number of cases, while it is also necessary to look for mutations in other moderate and low penetrance genes. Of these, genes which are crucial in DNA repair and DNA damage response have been postulated as good candidates.

The main goal of this work was to detect mutations in other genes to explain inherited breast/ovarian cancer predisposition, enabling the clinical management of the carrier families. Firstly, we focused on some FA/BRCA pathway genes such as *BRIP1, ERCC4*, *PALB2*, *RAD51C* and *RAD51D.* Since *ATM* has a pivotal role in the detection and response to DNA damage, and *BARD1* binds *BRCA1* mediating the initial nucleolytic resection of DNA lesions and the recruitment of *RAD51*, we decided to examine these key DNA damage response genes as well. The FA genes were screened by heteroduplex analysis by capillary array electrophoresis (HA-CAE) [13] or high resolution melting analysis (HRMA) as these techniques were used in our routine until 2018. Thereafter, we adopted NGS thus we decided to test *ATM* and *BARD1* with this technology. Our results demonstrate that increasing the number of assessed candidate genes, molecular diagnosis improves.

*ATM* had been known to be linked to breast cancer predisposition in BRCA-negative families [14,15]. Furthermore, *ATM* mutations have been found in ovarian cancer [16]. In accordance with this, we identified three *ATM* pathogenic variants in the index-case with breast cancer, one of whom also developed an ovarian cancer (Table 1). The patient II.5 (Figure 1C), diagnosed with breast cancer at the age of 43, carried the *ATM* variant c.3663G > A. Her family had, on the paternal side, two very early cancer cases at 25 (I.10) and 36 (I.11) years-old (Figure 1C). Unfortunately, DNA was not available for segregation analysis.

The *ATM* c.8934_8935delTG variant was found in the III.12 breast cancer patient (Figure 1d). Her daughter, IV.2, who carries the mutation, may benefit from prevention measures [2,3]. This mutation was previously reported in an Ataxia-Telangiectasia family [17]. In addition, other cancer types in this family have been associated with *ATM* mutations such as thyroid [18], ovarian [16] and breast tumours [15].

The variant c.4776+2T>C in *ATM* (Figure 4), previously reported in a breast cancer case [19], modifies a canonical splicing site, potentially resulting in a splicing disruption. In addition, cells from an individual, homozygous for this mutation, showed an increased radio-sensitivity [12]. This mutation was identified in a triple-negative breast cancer patient at 52 years-old and ovarian cancer at the age of 60. It is worth noting that several cancer types associated with pathogenic *ATM* mutations were all present in her family (Table 1): three gastric cancer cases [20], two breast cancers [19], two pancreatic cancers [21] and three colon cancers [22]. To assess the pathogenic effect of this mutation, we have experimentally demonstrated that this genetic change causes an alteration of splicing (Figure 4). Although this variant causes an in-frame deletion of 55 amino acid, it has been classified as pathogenic (ClinVar, and ACGM guidelines).

The implication of pathogenic *BRIP1* variants in ovarian cancer predisposition have been described before [16,23]. The c.484C > T *BRIP1* mutation was identified in an ovarian and melanoma cancer patient (Figure 1a, II.12). Interestingly, the mutation was inherited by her daughter, who developed breast cancer at age 32 (III.2).

Two *ERCC4* loss of function variants have been identified in two breast cancer cases. The c.584+1G>A mutation has been previously published as part of a Spanish collaborative study [11]. The patient has a strong breast cancer family history, with five affected women (Table 1). Unfortunately, DNA of the other breast cancer cases was not available to confirm familial segregation. In fact, three out of the five affected women died immediately after being diagnosed, so we assume tumour aggressiveness. The other *ERCC4* mutation—c.1251T>A—was identified in a bilateral breast cancer case (Table 1). Three endometrial cancer cases (the mother and two sisters of the patient) and two gastric tumours are the family antecedents. Other authors have associated *ERCC4* mutations with endometrial cancer [24,25] and colorectal [26]; however, until now, the relatives with endometrial cancer have refused genetic testing. Both mutations are classified as pathogenic according to the ACMG standards.

Only one pathogenic variant in the *RAD51D* gene- c.94_95delGT- was detected in an ovarian cancer patient (Figure 1B individual III.1, Table 1). The association between *RAD51D* pathogenic mutations and ovarian cancer has already been established [23,27,28,29], endorsing the causal role of the c.94_95delGT *RAD51D* mutation in the patient’s pathogenesis. Furthermore, her son, III.3, carried the mutation; accordingly, the following information could be taken into account for prevention and follow-up: COSMIC database records *RAD51D* somatic mutations in gastric, prostatic, pancreatic and liver tumours. In fact, germline *RAD51D* variants and susceptibility to prostate cancer has also been described [30]. Clinical guidelines have been developed for some genes such as *ATM, BRIP1, PALB2* and *RAD51D* [2,3,31], enabling genetic counselling.

A large proportion of the variants identified in genetic analysis present conflicts of interpretation [32]; ambiguity concerning their molecular effect impairs clinical management. Therefore, it is essential to make an effort to classify these variants [33]. Hence, another goal of this work was to characterise the spectrum of rare variants, other than pathogenic ones, which may lead to an increased risk for breast/ovarian cancer predisposition. The CADD tool seems very useful for identifying causal mutations with a high score of pathogenicity [34,35]. Moreover, it offers a practical and unbiased approach for estimating the pathogenicity of human genetic variants by integrating many diverse annotations into a single, quantitative score [35].

After CADD analysis, a total of 14 missense variants with high-CADD score were selected, apart from the eight properly pathogenic mutations. Interestingly, two variants altering the splicing, c.584+1G>A in *ERCC4* and c.4776+2T>C in *ATM*, presented a high-CADD score, denoting a possible link between CADD output and cDNA analysis.

In practice, the CADD score alone could not be used for variant classification, but the algorithm might be useful for prioritising variants for further functional and segregation studies [35].

Current records of the response to certain drugs, depending on the genetic profile, can serve as a reference [36] for the chemotherapy regimen choice. The main challenge in the genetic diagnosis of hereditary cancer is to transfer the findings to clinical practice; integrating information from in silico tools and drug response records to allow us to make these mutations useful.

In a pharmacological context, the Cancer Genome Interpreter compiles information about the implications of these possible causal variants for drug response. It is worth noting that most of the variants with a high-CADD score have been catalogued as driver mutations at the somatic level, assuming that they confer an advantage to the cell, triggering the transformation. Regarding the therapeutic approach, these relevant mutations may be used for drug selection: for *PALB2* mutations, a response of the tumours has been registered for PARP inhibitors [37,38], mitomycin C [39] and platinum compounds [40]. for deleterious *ERCC4* variants, the effectiveness of cisplatin is expected for disrupting mutations [41]; PARP inhibitors are selectively toxic to tumours with *RAD51C* mutations [42], and tumours with *ATM* mutations had a positive response to cisplatin [43], and inhibitors of PARP1 [37], ATR [44,45] and DNA-PKc [46].

Tumour onset in hereditary cancer is expected to be a distinguishing feature from sporadic cases: germline mutation would imply a high risk for early-onset cancer. According to this hypothesis, we checked whether the mean age of diagnosis was different between high-CADD score mutation carriers and non-carriers. Statistically, we could not discard the fact that the diagnosis age was the same in the two compared groups. This could be due to sporadic tumours that appear earlier as a consequence of other risk factors such as lifestyle. As a consequence, the inclusion of sporadic cancer cases in our cohort is possible. The immediate impact of the inclusion of sporadic cases is the presence of samples without genetic variants, which, at the same time, decrease the mean age of this group [47,48].

Conversely, a trend was observed upon close examination of the family information: carrying a pathogenic or rare variants with high-CADD score mutation seems to be determined by the presence of a family history with more cases and more diverse tumour types. Interestingly, the statistical analysis confirmed this association between high-CADD score mutation carriers and family cancer history: patients with these mutations presented more cancer cases and more tumour types in their pedigrees. This supports the hypothesis that familial history is relevant in sample selection for cancer predisposition [49]. In brief, we may face a possibly non-typified hereditary cancer syndrome, in view of the family history profile of our positive families (tumour type diversity beyond breast and ovarian cancer). In fact, different cancer syndromes have overlapping clinical features, so expanding the analysed gene set might imply a higher diagnostic yield.

## 4. Materials and Methods

### 4.1. Patients

The selected 180 cases were screened for deleterious mutations in the BRCA1 and BRCA2 genes, and resulting negatives. We have studied 180 probands from breast and ovarian cancer families meeting high-risk criteria, enrolled from the Regional Hereditary Cancer Prevention Program of Castilla-León (Spain). The participants in this study are ovarian or breast cancer patients who have first-degree ovarian cancer relatives in the same side of the family. Ethical approval for this study was obtained from Comité Ético de Investigación Clínica (CEIC). Informed consent, family history and clinical features were collected (The approval code from the ethical committee is PI-13-66 and it was obtained at the meeting of the Clinical Committee on 03/21/2013).

### 4.2. DNA and RNA Extraction

Genomic DNA from peripheral blood was automatically extracted by Roche MagNaPure^®^ Compact, using the “MagNA Pure Compact Nucleic Acid Isolation Kit I—Large Volume” (Roche Diagnostics, Penzberg, Germany), following the manufacturer’s instructions. RNA was extracted from peripheral blood lymphocytes using the GeneMATRIX Human Blood RNA Purification Kit (EURx, Gdánsk, Poland). DNA concentration was measured in a NanoDrop™ (NanoDrop 2000c, ThermoFisher Scientific, Waltham, MA, USA).

### 4.3. Mutation Screening

The entire coding sequence and splicing sites of *PALB2* and *ERCC4* were screened using HA-CAE [13]. *RAD51C*, *RAD51D* and *BRIP1* were screened by HRMA. 

*RAD51C*, *RAD51D* and *BRIP1* specific primers were designed with the UMelt software [50] Ideally, amplicons ranged from 100–250 bp with a single melting peak. Our design targeted the coding exons and the corresponding exon/intron boundaries. Sequences of primers are available upon request.

PCR (HRMA were conducted in a LightCycler 480 Instrument (Roche Diagnostics, Penzberg, Germany). All reactions were performed in 10 μL final volume. PCR annealing temperatures (Ta) and MgCl_2_ concentrations were optimised for each amplicon, lastly confirmed by direct sequencing. In brief, for PCR reactions, we used the LightCycler^®^ 480 High-Resolution Melting Master Mix kit (Ref: 04909631001, Roche): Mix HRM Roche 1×, 2.5 mM MgCl_2_, (3.5 mM of MgCl_2_ when the amplification was suboptimal), 2.5 μM each forward and reverse primers and 2 ng/μL of genomic DNA. Thermal cycling consisted of an initial 10-minute hold at 95 °C, followed by 10 s hold at 95 °C, 15 s hold at the specific Ta, and 20 s hold at 72 °C for 40 cycles. Consecutively, HRMA consisted in a 65 °C to 95 °C melting gradient with a 0.02 °C/s ramp rate and continuous 25 acquisitions/°C mode. The LightCycler 480 Software version 1.5 (Roche Diagnostics, Indianapolis, IN, USA) was used for the melting curve analysis. After checking the melting curves, we selected the temperature to normalise data. When the curves differed in shape and/or melting temperature, the corresponding samples were subsequently sequenced. Primers were designed using Primer3 tool software (http://bioinfo.ut.ee/primer3-0.4.0/).

*ATM* and *BARD1* were sequenced using MiSeq Technology at the Centre for Medical Genetics Ghent, as previously described [51].

Mutation nomenclature was based on the following reference sequences: *ATM* NM_000051.3; *BARD1* NM_000465.3; *BRIP1* NM_032043.2; *ERCC4* NM_005236.2; *PALB2* NM_024675.3; *RAD51C* NM_058216.1; *RAD51D* NM_002878.3.

### 4.4. Sanger Sequencing

Direct automated Sanger sequencing was used to confirm the results detected by screening methods and massive parallel sequencing. For that purpose, we used the BigDye Terminator Sequencing Kit v3.1 (Applied Biosystems, Foster City, CA, USA) on an ABI 3100 DNA Sequencer (four capillaries; Applied Biosystems). Co-segregation studies were conducted when possible.

### 4.5. RT-PCR

Variants predicted as disrupters of splicing in the Human Splicing Finder TM 3.0 (HSF) software (http://www.umd.be/HSF3/) were selected to perform cDNA-based analysis. Total RNA isolated from lymphocytes was reverse transcribed into cDNA using the Transcriptor First Strand cDNA Synthesis Kit (Roche), according to the manufacturer’s protocol. Subsequently, we performed a PCR to evaluate the transcription. The products were separated in low melting 2% agarose gel and visualised with RedSafe™ nucleic acid staining solution (iNtRON Biotechnology, Korea) Isolated bands were extracted using NucleoSpin^®^ Gel and PCR Clean-up (Macherey-Nagel, Düren, Germany) and subsequently sequenced.

We designed the primer pair −29 Forward (5′-TGTGAGCAAGCAGCTGAAACAA-3′) and 33 Reverse (5′- TTCACCAGTGTGGTTTATTGCCA-3′)— to evaluate the transcript spanning exons 29–33 so as to characterise the splicing effect of the c.4776+2T>C *ATM* variant. The RT-PCR reaction consisted of a 1X Buffer A, 0.5  µM forward and reverse primers, 0.32  mM dNTP Mix, 1 Unit of Kappa Taq DNA Polymerase, 12  µL of the cDNA generated in a final volume of 100  µL. The cycling conditions were denaturation at 95 °C for 3 min, 35 cycles at 95 °C for 30 s, 62 °C for 30 s, and 72 °C for 50 s, followed by a final extension at 72 °C for 10 min.

### 4.6. In-Silico Analyses

Mutations with protein annotations were analysed using the MutationMapper (http://www.cbioportal.org/mutation_mapper), the cancer genome interpreter (https://www.cancergenomeinterpreter.org) and HSF. Mutations that lead to premature truncation of the protein, frameshift, nonsense, and splice site, were classified as deleterious mutation. On the other hand, missense variants with minor allele frequency (MAF) <0.01 according to ExAC data were selected to further in silico research (hereinafter rare variants). Combined Annotation-dependent Depletion (CADD) was applied for both deleterious mutations and rare variants. In the CADD method, the scaled- C-scores are related with the top-ranked pathogenicity: to CADD-Score-10 you are in the top 10% of the disrupting mutations, to CADD-Score-20, top 1%, CADD-Score-25, top 0.5% and to CADD-Score-30, 0.1%. CADD integrates diverse annotations into a single score by contrasting variants that survived natural selection with simulated mutations [35,52,53]. We considered variants with a CADD-score of >25 as possibly pathogenic (top 0.5% of disrupting variants). 

All data representations, including the circular Barplot (Figure 3) and Heatmap (Figure 5), were made using the Complex Heatmap (https://doi.org/10.1093/bioinformatics/btw313) and ggplot2 [54] packages with the R Project for Statistical Computing, v.3.5.1. (https://www.r-project.org.)

### 4.7. Variant Classification

Variants were classified as deleterious if they originated a premature stop codon, if they were located in canonical splice sites, or if there was literature evidence (ACGM guidelines, ENIGMA classification rules). The potential deleteriousness of the remaining rare variants was evaluated using the CADD method.

### 4.8. Genotype-Phenotypic Correlations and Statistical Analysis

Personal and family data and mutation profiling of the samples were compiled and explored, looking for genotype–phenotypic correlations. Statistical tests were carried out using the R Project for Statistical Computing (v.3.5.1). A one-sided t-test was used for two-group means comparison, and Pearson’s chi-squared test with Yates’ continuity correction was used to evaluate the association between two categorical variables.

## 5. Conclusions

A large proportion of breast and ovarian cancer predisposition remains unexplained: analysis of genes other than *BRCA1* and *BRCA2* is needed. We have contributed to the molecular diagnostics of some HBOC families in whom previously BRCA1/2 was ruled out, analysing other low to moderately penetrant genes. The detection of deleterious mutations and other prioritised variants in some of these genes has a significant impact on clinical management. Nevertheless, further studies are needed to enable prevention, early detection and treatment selection in HBOC. Our data suggest that carriers of variants affecting the function of the gene belong to families with more cancer cases and a greater diversity of tumour types.

## Figures and Tables

**Figure 1 cancers-12-02151-f001:**
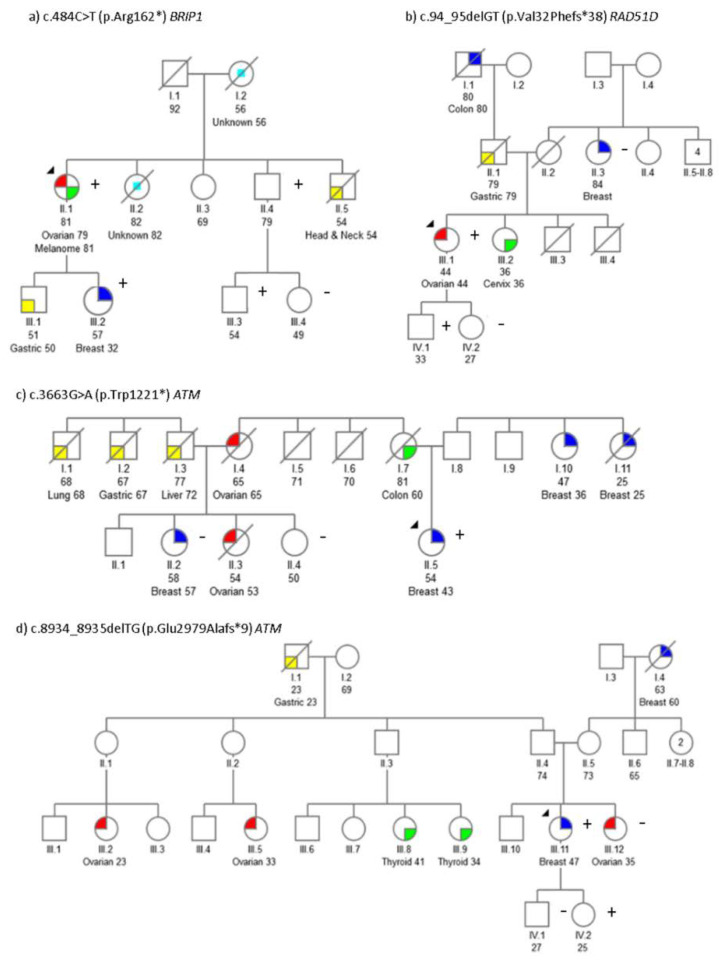
Segregation of *BRIP1*, *RAD51D* and *ATM* pathogenic mutations. (**a**) Family with c.484C>T (p.Arg162*) *BRIP1* mutation, (**b**) Family with c.94_95delGT (p.Val32Phefs*38) *RAD51D* mutation, (**c**) Family with c.3663G>A (p.Trp1221*) *ATM* variant; (**d**) Family with c.8934_8935delTG (p.Glu2979Alafs*9) variant in *ATM*. Index cases are indicated by an arrow. Confirmed mutation carriers are indicated by a “+” sign and non-carriers by a “−” sign. Age at diagnosis and cancer type is specified when known. Ov (ovarian), Mel (melanoma), Gas (gastric), Thy (thyroid), Liv (liver), Lung (lung), Br (breast), H&N (head and neck), Col (colon), Cvx (cervix) and CUP (unknown primary tumour).

**Figure 2 cancers-12-02151-f002:**
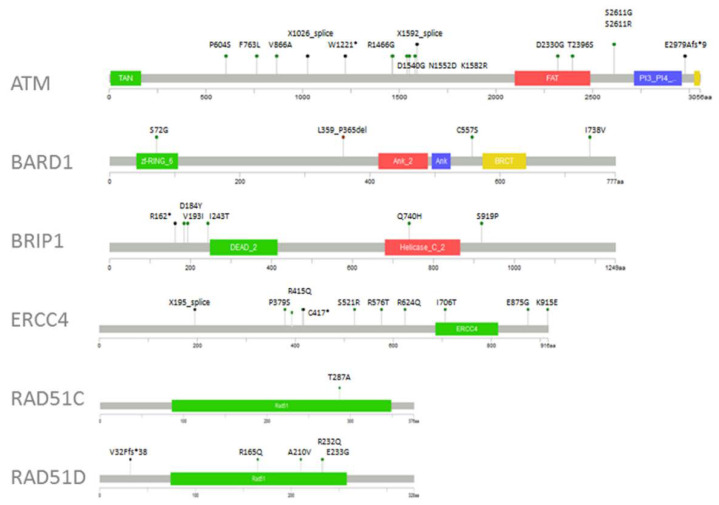
Potentially damaging missense variants. Pathogenic and missense variants with possible functional impact are presented above the corresponding protein scheme (MutationMapper tool from cBioPortal for Cancer Genomics). Most of the variants were listed in the ExAC database (Exome Aggregation Consortium, non-Finnish Europeans) with minor allele frequency (MAF) < 0.01.

**Figure 3 cancers-12-02151-f003:**
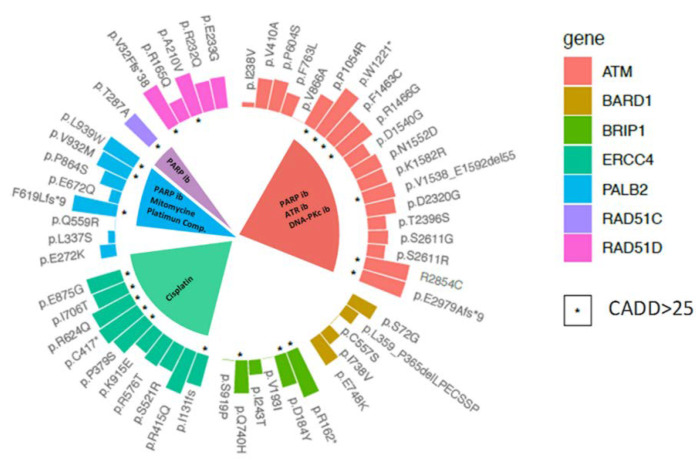
Circular barplot showing the distribution of CADD scores for different variants per gene. Variants with CADD score > 25 are indicated by “*”. Apart from the loss of function protein variants, other rare missense mutations were prioritised according to CADD-score. For *ATM*, p.Pro1054Arg, p.Phe1463Cys, p.Arg1466Gly and p.Arg2654Cys; for *BRIP1*, p.Asp184Tyr and p.Gln740His; for *ERCC4* p.Pro379Ser, p.Arg624Gln, p.Ile706Thr and p.Glu875Gly; for PALB2 p.Val932Met and p.Lys939Trp; for *RAD51C* p.Thr287Ala; for *RAD51D* p.Ala210Val. Different drugs for which a response is predicted for some variants with a CADD score >25 in several genes are indicated in the inner part of the circular barplot. “ib” inhibitors.

**Figure 4 cancers-12-02151-f004:**
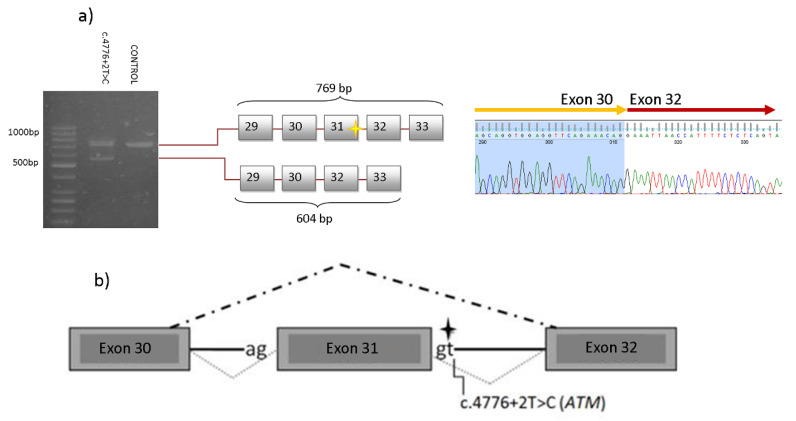
Characterization of the c.4776+2T>C variant of *ATM*. (**a**) PCR products were separated by a 2% agarose gel electrophoresis. The c.4776+2T>C variant caused the skipping of exon 31, resulting in a product which is 165 nucleotides shorter than the full-length product (769 nucleotides). (**b**) Schematic representation of the exon skipping caused by c.4776+2T>C *ATM* variant.

**Figure 5 cancers-12-02151-f005:**
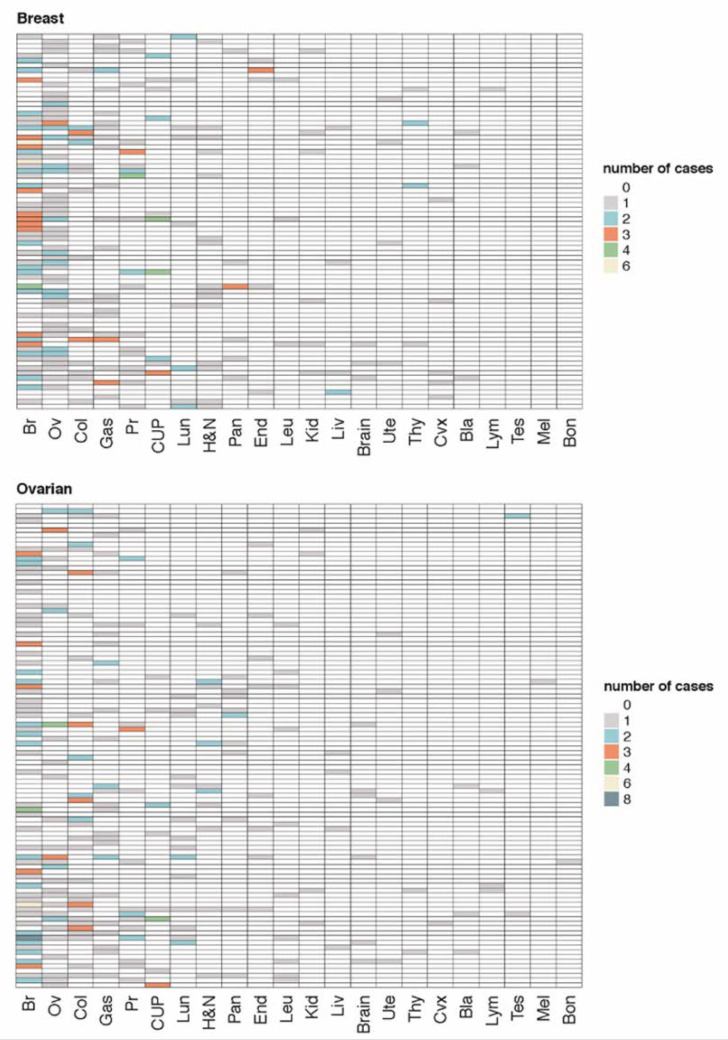
Personal and family cancer history of the different studied index cases. Index cases are clustered according to their personal cancer history, such as "breast" or "ovarian" when they developed ovarian or breast cancer, respectively. Each row represents the family pedigree of the index cases and the columns represent the different types of tumours in the family where each colour indicates the number of cases for a specific type of cancer. Col, colon; Gas, gastric; Pr, prostatic; CUP, cancer of unknown primary origin; Lun, lung; H&N, head and neck; Pan, pancreatic; End, endometrial; Leu, leukaemia; Kid, kidney; Liv, liver; Brain, brain; Ute, uterus; Thy, thyroid; Cvx, cervix; Bla, bladder; Lym, lymphoma; Tes, testicular; Mel, melanoma; and Bon, bone cancer.

**Table 1 cancers-12-02151-t001:** Pathogenic variants identified in the 180 index patients.

Gene	c.DNA Change	Protein Change	ACMG Rules	Proband Cancer Type (Age)	Family History Cancer Type (Age)		
					First-Degree-Relatives	Second-Degree-Relatives	Third-Degree-Relatives
*ATM*	c.3663G>A	p.Trp1221 *	PVS1, PM2	Br (43)	Col (50)	Br (36), Br (25), Ov (62)	Br (57), Ov (53)
*ATM*	c.4776+2T>C	-	PVS1, PM2	Br (52) Ov (60)	Pan (29), Pan (58), Br (47), Col (67), Col + Br (66,74)	CUP (78)	Gas (58), Gas (65), Gas (70), Gas (72), Br + Col (66,74)
*ATM*	c.8934_8935delTG	p.Glu2979Alafs*9	PVS1, PM2	Br (47)	Ov (35)	Gas (23), Br (60)	Ov (23), Ov (33), Thy (34), Thy (41)
*BRIP1*	c.484C>T	p.Arg162 *	PVS1, PM1, PM2	Ov (79)Mel (81)	Gas (50), Br (32), H&N (54), CUP (82), CUP (56)		
*ERCC4*	c.1251T>A	p.Cys417 *	PVS1, PM1, PM2	BilBr (59-59)	End (47), End (51), End + Col (57,59), Gas (80)	Br (70), Gas (75)	Br (72)
*ERCC4*	c.584+1G >A	-	PVS1, PM2	Br (53)	Br (55)	Col (45), Br (54), Br (33), Br (57), Pr (84)	Col (36), Ute (47), Br (53), Br (55)
*PALB2*	c.1857delT	p.Phe619Leufs*9	PVS1, PM1, PM2	Ov (33)	Br (36), Liv (78)	End (55), Lun (62)	Leu (n.a)
*RAD51D*	c.94_95delGT	p.Val32Phefs*38	PVS1, PM1, PM2	Ov (44)	Cvx (36), Gas (79)	BilBr (84)	Col (80)

Cancer types are abbreviated as following: Br, breast; BilBr, bilateral breast; Ov, ovarian; Gas, gastric; Col, colon; Pan, Pancreas; Thy, thyroid; H&N, head and neck; Pr, prostrate; Cvx, cervix; Ute, uterus; Kid, kidney; CUP, cancer of unknown primary origin. The age of diagnosis is indicated in parentheses, n.a not available data. * nonsense mutations; the nucleotide substitution introduce premature termination codon.

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
