# Peer review of "Germline Genetic Findings Which May Impact Therapeutic Decisions in Families with a Presumed Predisposition for Hereditary Breast and Ovarian Cancer"

_cancers, 2020, doi:10.3390/cancers12082151_

Round 1

Reviewer 1 Report

Comment 1

As proven in the authors' investigation, depending on the pathogenic level of each ATM mutation, each ATM mutation's penetrance level may be severely different regarding to risk for BRCA1/2-negative breast cancer or other cancer types. Therefore, as in line 194 (in the author's manuscript), considering ATM gene as a moderate-penetrance gene is inappropriate. Please, change the line 194 sentence description as follows: 

ATM had been known to be linked to breast cancer predisposition in BRCA-negative families [12,13].  

Author Response

We have changed the line 194 sentence description, (now in line 212) as the suggestion “ATM had been known to be linked to breast cancer predisposition in BRCA-negative families [12,13].”

Reviewer 2 Report

Velázquez et al., studied to gain insight in the germline mutation spectrum of ATM, BARD1, BRIP1, ERCC4, PALB2, RAD51C and RAD51D in breast and ovarian cancer families from Spain. The study is done for 180 index cases not mutated in BRCA1 and BRCA2 in germline. Using CADD as a major tool, the authors are scored those mutations induced in germ line. Conceptually, I found some potential interest in the study. However, some additional studies need to be done.

Major points

1: Although the authors claimed some “therapeutic implications”. This is only speculation. This must be done for at least for some to check if their concept is really the case, or not.

In Lines 118-120, the authors stated that “A CADD score >25 was previously suggested as cut-off for of a damaging effect. Indeed, all pathogenic mutations surpassed that CADD score.” This might be. But this should be somehow done.

In Section 2.4 Therapeutic implications, this section is completely a speculation. This should be experimentally shown.

2: I hardly understand Figure 5. I can imagine many people would have same problem with me. But I do not understand how to look Personal and family cancer history in this figure. What exactly each horizontal line corresponds?

3: One of the critical questions is if those mutations studied in this manuscript are similar to the mutations of BRCA1 and 2. Are any similarities or the differences observed

Author Response

Answer to point 1:

We appreciate the suggestions of reviewer 2 on the “therapeutic implications” section. We totally agree that the selection of therapy according to the mutational profile should be experimentally proved but, in an attempt to make the data obtained in the mutational screening useful, we have used the compiled information on the response to treatment linked to some genetic mutations in cancer cases. Although ideally each particular mutation should be evaluated for the different drug effects, which implies in vitro and in vivo studies to uncover the specific response, this is not feasible in most of the cases. As an alternative, and since the NGS is implemented in clinics, we wanted to use the registered responses of many cancer patients to different drugs based on the mutations found in the tumour. Since a wide variety of targeted inhibitors can be offered to patients, we found it interesting to share with the scientific community the assumption of therapy selection related to mutational results. Nonetheless, we have rephrased the paragraph to underline the hypothetical connotation of the stated premise (lines 156-162 of the corrected manuscript).

We hope that the part concerning the therapeutic implications will now be perceived as a reasoned but inconclusive information that, undoubtedly, will require further research.

Answer to point 2:

We regret the poor display of the cancer history in the heat map representation. To solve this, we have redrawn the graph to clearly show the information on the different breast and ovarian cancer index cases and the respective familial cancer cases. Moreover, we have made the figure 5 caption much more explanatory.

Answer to point 3:

We find this commentary very interesting. The patients in whom we found any pathogenic mutation in the screened genes were previously analysed for BRCA1 and BRCA2. This means that they were selected following the same guidelines and, therefore, the criteria as a family history, age of diagnoses or cancer type are the same. Due to BRCA1 and BRCA2 are well-studied genes and they have long been analysed in Hereditary Breast and Ovarian cancer cases, the information about the penetrance and risk is more accurate than for the other genes. The implementation of NGS in the genetic screening will allow us to select more genes to add and complete information about their implication in cancer predisposition. In particular, as it is described in the literature the majority of PALB2-associated breast cancers are ER positive, such as those mutated with BRCA2 (Macedo GS et al. Genet Mol Biol. 2019;42(1 suppl 1):215-231. doi:10.1590/1678-4685-GMB-2018-0104).

Reviewer 3 Report

This article demonstrates the importance of panel genetic testing, which has largely been widely adopted. However it is not clear what is standard practice in Spain. Recommend further elaboration of current practice in area to frame the relevance of this study. 

Author Response

We appreciate the reviewer's comments. In Spain, particularly in our region, the standard practice for patients at high risk for HBOC is to screen germline mutations in BRCA1 and BRCA2 genes. With NGS technology we can analyse a large number of genes in a single run. The aim of this work was a first approach to select clinically valuable genes useful for management in our BRCA-negative families. These families with a previous negative result took profit from prevention measures described in the SEOM clinical guidelines (reference 3). Furthermore, the results of this work lead us to propose a modification in our clinical routine and implement a custom panel for families that meet high-risk criteria in our target population.

Thus we have changed add some sentences in lines 70-77 of the corrected manuscript.

Reviewer 4 Report

I've read with interest the paper titled "Germline genetic findings which may impact therapeutic decisions in families with a presumed predisposition for Hereditary Breast and Ovarian Cancer", corresponding author Mar Infante.

The paper covers an interesting topic since the prevalence of other germline mutations is of interest in such population. 

I think the manuscript explanation is quite straightforward, I think it is worth an acceptance, however, I have some comments to improve the paper:

*did you test the same mutations in the tumor? It'd be of interest to identify the tumor heterogeneity and conservation of the mutation.

*Line 78 Why you chose this group of genes leaving others outside?
*Line 90 "pancreas" is missing in the Legend

*Some more details are required about the CADD Score, little reference provided

*line 190 no definition was found for HA-CAE, HRMA please state
*agarose gel figure 4.a is dark difficult to interpret

Author Response

*did you test the same mutations in the tumor? It'd be of interest to identify the tumor heterogeneity and conservation of the mutation.

Unfortunately, we only receive blood samples from patients. Certainly, it would be really informative evaluate the presence of the germline alterations in the tumour. We have scheduled to do so in the near future; but, at the moment, it is not possible.

*Line 78 Why you chose this group of genes leaving others outside?

We have rewritten the paragraph to better explain why we chose these group of genes (lines 83-85 the corrected manuscript). The reasons are better explained in discussion (lines 203-207 of the corrected manuscript). After studying BRCAs genes, we focused on FA-BRCA genes, so we analysed BRIP1, ERCC4, PALB2, RAD51C and RAD51D with the techniques used in the laboratory at that time. Thereafter, we adopted NGS thus we decided to test ATM and BARD1 with this technology.

*Line 90 "pancreas" is missing in the Legend

We have added Pancreas to the legend (line 96 of the corrected manuscript)

*Some more details are required about the CADD Score, little reference provided.

We have included two more references, were CADD has been used for Score deleteriousness variants (line 359 of the corrected manuscript)

*line 190 no definition was found for HA-CAE, HRMA please state.

HA-CAE, HRMA were defined later in Material and Methods section 4.3. We have moved the definitions to the right place, now they are in line 207 and 208 of the corrected manuscript.  

*agarose gel figure 4.a is dark difficult to interpret.

We have improved the image of the agarose gel.

Round 2

Reviewer 2 Report

Velázquez et al., revised the manuscript entitled “Germline genetic findings which may impact therapeutic decisions in families with a presumed predisposition for Hereditary Breast and Ovarian Cancer”. The revised manuscript is better.

However, Figure 5 is still not well improved. Although number of cases were illustrated with gray scale, this should be changed to heatmap scale. Otherwise it is hard to see.

I have no further comment on this manuscript.

Author Response

We agree with the reviewer. To increase readability and aid interpretation, we have changed the greyscale to a colour one. Furthermore, we have moved from a continuous scale to a discrete one where each colour represents a specific number of cases in the family.

Still, the upper block represents families where the index case was breast cancer whereas the bottom panel represents families where the index case was ovarian cancer.

Reviewer 3 Report

No further suggestions

Author Response

English language has been checked by a native British speaker.